# The Strong Coupling Effect between Metallic Split-Ring Resonators and Molecular Vibrations in Polymethyl Methacrylate

**DOI:** 10.3390/s24082479

**Published:** 2024-04-12

**Authors:** Ya Liu, Esha Maqbool, Zhanghua Han

**Affiliations:** Shandong Provincial Key Laboratory of Optics and Photonic Devices, Center of Light Manipulation and Applications, School of Physics and Electronics, Shandong Normal University, Jinan 250358, Chinaishach3137@gmail.com (E.M.)

**Keywords:** split-ring resonators, strong coupling, plasmon resonance, vibrational resonance

## Abstract

We propose and study a nanoscale strong coupling effect between metamaterials and polymer molecular vibrations using metallic split-ring resonators (SRRs). Specifically, we first provided a numerical investigation of the SRR design, which was followed by an experimental demonstration of strong coupling between mid-infrared magnetic dipole resonance supported by the SRRs fabricated on a calcium fluoride substrate and polymethyl methacrylate molecular vibrations at 1730 cm^−1^. Characterized by the anti-crossing feature and spectral splitting behaviors in the transmission spectra, these results demonstrate efficient nanoscale manipulation of light–matter interactions between phonon vibrations and metamaterials.

## 1. Introduction

The majority of naturally occurring materials, consisting of molecules and atoms, offer limited control over their physical properties. Fortunately, the emergence of metamaterials can circumvent this limitation by providing unprecedented flexibility in tailoring the material properties. The metamaterials are kinds of artificial structures which are carefully designed and constructed. Through the precise control of their meta-atoms, unique physical characteristics beyond natural materials can be achieved, which are hard to access in naturally available materials. The emergence of metamaterials has revolutionized people’s control over our world and opened up new possibilities to develop high-performance devices and technologies. By a fine adjustment of key parameters of the meta-atoms, such as the permittivity and permeability, topological structure, and size, metamaterials can exhibit rich and diverse physical properties. For example, metamaterials can achieve a near-zero dielectric constant [1], which implies that their phase response to the electromagnetic waves is negligible at some frequencies, providing enhanced control over the manipulation and propagation of electromagnetic waves. The perfect absorption [2] offered by metamaterials provides strong prospects for stealth technology, energy conversion, and other fields. Those metamaterials that support strong resonances, e.g., localized surface plasmon resonance (LSPR) or Mie resonances, have an extraordinary ability to concentrate the electromagnetic field in the subwavelength range, thus greatly enhancing the near field. Due to their distinctive characteristics, such metamaterials have broad application prospects for biological detection. One famous example is surface-enhanced Raman scattering (SERS) technology [3], in which one can effectively improve the Raman-scattering signal intensity by using such metamaterials, and then realize a high sensitivity and high specificity in the detection of biomolecules.

The resonating nature of the meta-atoms can lead to complex and interesting phenomena, such as Fano resonance [4] and electromagnetically induced transparency (EIT) [5], among which the strong coupling phenomena [6] between an optical resonance and a material excitation (such as an exciton or molecular vibration) is of great importance in both fundamental research and technological applications. The necessary condition for a strong coupling to occur is that the energy exchange rate between the optical resonance and the material excitation should be much higher than the intrinsic dissipation rate in either of the two constituent states. This condition ensures that desired interaction between light and matter is strong enough to overcome the internal losses of the system and realize an efficient energy transfer. At the occurrence of the strong coupling effect, two new polariton states will be generated which inherit the hybrid properties of photons and matter. As a result, when strong coupling occurs, the properties of the original optical resonance and matter are retained while the generated new polaritons [7] exhibit significantly different physical and even chemical responses. The important influence of the strong coupling effect is quite diverse. In terms of basic research, the light–matter strong coupling serves as a key experimental platform for exploring quantum electrodynamics and condensed-matter-physics phenomena. By interacting light fields with excited states of matter, new energy structures and collective behaviors can be generated, such as Bose–Einstein condensates [8]. This not only contributes to an in-depth knowledge of the key mechanisms by which light interacts with matter, but also provides a potential basis for novel quantum devices and quantum information processing. In terms of technical applications, light–matter strong coupling can be used to develop high-performance optical devices. In addition, the development of optical sensing technology also benefits for the regulation of such interactions. It uses the signal changes generated by the interaction between matter and light to detect and identify targets, providing accurate and efficient solutions for environmental monitoring, biomedicine, and other fields. In the field of optoelectronics, light–matter strong coupling can also be utilized to realize ultrafast light modulator components [9] and efficient photovoltaic materials [10]. In the meantime, in the field of biomedicine, light–matter strong coupling is also widely used in fluorescence imaging, photothermal therapy, and other aspects [11]. Therefore, the light–matter strong coupling holds great significance in the development of novel optical technologies along with the promotion of great scientific research.

Due to the large abundance of molecular-vibration modes in the mid-infrared (MIR), the study of the strong coupling between metamaterials and molecular vibrations has attracted ever increasing attention in recent years. Shelton et al. used split-ring resonators (SRRs) to explore the strong coupling mechanism between planar metamaterials and infrared (IR) active phonon vibrations of silicon dioxide in nanoscale dielectric layers [12]. They successfully achieved the strong coupling phenomenon by carefully adjusting the size of the SRRs. However, their results require the preparation of many samples with different meta-atom sizes, which not only complicates the fabrication process, but also requires quite accurate control over the meta-atom size. MAHMUD et al. successfully realized the strong coupling phenomenon between a chiral metasurface and polymethyl methacrylate (PMMA) under the excitation of circularly polarized light [13]. Ma and colleagues used the “U” metal structure on the epsilon-near-zero (ENZ) substrate to successfully realize the strong coupling with the molecular vibration of PMMA molecules [14]. Through in-depth experimental research and data analysis, they successfully verified the linear relationship between the coupling strength and PMMA thickness, which provided a solid basis for our subsequent study of the PMMA molecules interactions with optical resonators.

Although the strong coupling effect is fundamentally appealing, its realization using regular structures still remains elusive. Besides the above-mentioned experimental demonstration, it has also been reported in previous studies that microcavity mode [15,16], metal gratings [17,18] or thin metal films [19] can help achieve the strong coupling effect. However, these resonators have some limitations in controlling the mode volume [20], which affects the coupling strength between electromagnetic waves and matter. For the application in sensing, there is the need for an easy-to-fabricate structure whose resonance is also highly sensitive to the introduced target material in the surroundings. In this respect, the SRR, offering advantages including structural simplicity, convenient fabrication, and desirable performance, still represents a good candidate for the realization of the strong coupling effect. The resonant frequency and local field enhancement effect can be controlled by adjusting the size, shape, and structure of the SRR. Actually, the SRR structure has been widely used in micro- and nano-optics fields. It has not only been used as an efficient absorber [21,22], which can accurately control the absorption process of light [23], but also has played an important role as a filter [24,25,26] to effectively screen out light waves at specific frequencies. Lahiri et al. investigated an asymmetric SRR and obtained a 0.51 µm and 0.12 µm spectral shift for a 100 nm and 30 nm thick PMMA layer [27], respectively. Ma et al., studied the deposition of a PMMA layer with a different thickness on gold SRRs and observed that a thickness of 300 nm is required to reach the vibrational strong-coupling regime. The highest spectral shift reported for 295 nm thickness was almost 0.13 µm [14]. Wan et al. demonstrated a cross-shaped SRR array milled into a 50 nm thick gold film and observed a strong coupling for different thicknesses of PMMA (40 nm, 100 nm, and 180 nm), but their highest spectral shift was limited to almost 0.5 µm [20]. Sun et al. studied the strong coupling between the bound state in the continuum modes supported by a germanium structure on CaF_2_ substrate and PMMA vibrations [28]. They observed through experiments that when the thickness of a PMMA layer is accurately controlled at 78 nm, its spectral shift is around 0.23 µm. Although these works have paved the foundation for the strong coupling in the MIR, the need for a new structure that shows a spectral shift for even smaller thickness of PMMA still remains to be addressed. That is important for immune sensing, security forensic analysis, and the sensitive sensing of small amounts of biochemical or chemical materials [27].

In this paper, we report an in-depth investigation through experiments and demonstrate that there is a high degree of sensitivity in the thickness of the PMMA layer introduced, which is particularly remarkable in the strong coupling between the SRR resonance and the vibration of molecules occurring in MIR band. Through the accurate experimental measurement of the transmission spectrum, we have observed a pronounced Rabi splitting phenomenon that is highly consistent with the finite element method (FEM) simulation results. Furthermore, we find that the calculated and measured spectral shifts are both highly sensitive to the PMMA-layer thickness, which fully demonstrates the effectiveness of our designed structure in sensing organic materials. This research not only deepens our understanding of the interaction between light and matter, but also provides strong support and new ideas for future applications in the field of organic-material sensing and optical-device design.

## 2. Structure and Results

Figure 1a illustrates a schematic representation of the SRR array under our investigation. Our designed structure is made from a 50 nm thick layer of silver (Ag) material located on a CaF_2_ substrate. This SRR array has the capability to regulate and manipulate electromagnetic fields in the MIR band. The spectral response or transmission and reflection spectrum can be used to provide a careful optimization of the SRR design parameters such as the period, size, and gap. In the MIR band, the dielectric constant, provided by the Drude model, can be used to describe how a metallic material responds to electromagnetic waves. For silver, the Drude dispersion model is given as
(1)$$\varepsilon_{Ag}=\varepsilon_{\infty }-{\omega_{p}}^{2}/(\omega^{2}+i\gamma_{p}\omega )$$
where *ε*_∞_ = 6.0 is a background dielectric constant, *ω* denotes the angular frequency of light, *ω_p_* denotes the plasma frequency, which describes the collective oscillation frequency of electrons in a metal, and *γ_p_* is the collision frequency, which describes the rate at which an electron energy is affected by the collision. These values of *ε_∞_ =* 6.0, *ω_p_ =* 1.5 × 10^16^ rad/s, and *γ_p_* = 7.73 × 10^13^ rad/s are used to characterize the Ag properties in the spectral band of our interest [29]. The refractive index of CaF_2_ can be expressed using the following formula [28]:(2)$$\varepsilon_{CaF_{2}}=1.33973+\frac{0.69913\lambda^{2}}{\lambda^{2}-0.09374^{2}}+\frac{0.11994\lambda^{2}}{\lambda^{2}-21.18^{2}}+\frac{4.35181\lambda^{2}}{\lambda^{2}-38.46}$$
where *λ* denotes the wavelength of incidence in a vacuum. The top view of a unit cell is depicted in Figure 1b, along with the subsequent geometrical parameters: the array period *P_x_* = 1.53 μm, *P_y_* = 2.2 μm, the radius *R*_1_ = 0.3*P_x_*, *R*_2_ = 0.185*P_x_*, the opening gap *g* = 0.1*P_x_*, and the width *w* = 0.1*P_x_*. These parameters give rise to a magnetic dipole (MD) resonance close to the PMMA molecular vibration around 1730 cm^−1^. To validate the MD response, a numerical investigation of the optical response of the SRR array was conducted using the commercial software COMSOL Multiphysics 6.0, which employs the finite-element method (FEM). Floquet periodic boundary conditions are used at the four lateral sides of the unit cell, thus mimicking an infinite planar metasurface structure, while the perfect matching layers (PML) are used in the *z* direction. A *y*-polarized plane wave is incident from the top to interact with the metasurface, and the transmission and reflection spectra from the structure can be retrieved from the S-parameters. In the simulations, the permittivity of materials with vibrational resonances of the molecular chemical bonds, such as PMMA, can often be characterized using the Lorentz model, which assumes that the interaction between electrons in a material and the light field is similar to the oscillation of a resonator. The model uses the following equation to describe the dielectric constant:(3)$$\varepsilon_{PMMA}=\varepsilon_{\infty }-\frac{f_{0}{\omega_{0}}^{2}}{\omega^{2}-{\omega_{0}}^{2}-i\gamma \omega }$$
where *ε_∞_* is the background relative permittivity of PMMA and its value is 2.2, *ω*_0_ is the Lorentz resonance frequency and its value is 3.253 × 10^14^ rad/s, the strength coefficient *ƒ*_0_ is 0.025, *γ* is the Lorentz damping rate, and its value is 3.0 × 10^12^ rad/s [28].

Figure 2a presents the computed transmission spectrum through the bare SRR array at a normal incidence. Our further results reveal that the resonance remains at the same frequency for inclined incidences. A significant resonance peak is observed at the wavenumber of 2037 cm^−1^ and the main factor preventing the resonator from reaching zero is the mismatch between losses from the silver absorption and from radiation leaking. The vectorial displacement current distribution across the *xy* cross-section at the resonance is depicted in Figure 2b. It is observed that the displacement current forms a sharp contrast between the left and right sides of the structure, with its flow direction completely opposite. This particular current-distribution pattern is fully consistent with the typical circulating characteristics of a MD [30]. Jie et al. utilized the MD resonance in a single silicon antenna and demonstrated significantly enhanced interactions between MIR light and nano-molecules, providing a strong proof for the contribution of MD resonance in MIR sensing [31]. For the resonance peak, the amplitude of the electric field is shown in Figure 2b. Remarkably, the amplitude of the electric field shown in Figure 2b can reach a maximum enhancement factor of 70 (the incident plane wave has an electric field of 1 V/m), which is a very promising result in the case of metals. Through detailed calculations, we found that the width of the arc-shaped part of the SRR is important to achieve the strong coupling between SRR and PMMA molecular vibrations. To highlight this point, in the next step, we decreased the value of *R*_2_ to 0.155*P_x_* and calculated the normalized scattering from individual SRRs instead. The scattering cross-section can be understood as the ratio between the number of photons scattered by the structure normalized to that in the incident light, and it can be calculated by the following [32]:(4)$$\sigma =\frac{1}{I}\int_{sphere}\frac{1}{2}\left(\vec{E}\times \vec{H}\right)\cdot \hat{n}dS$$
where $I=\frac{1}{2\eta }E_{0}^{2}$ stands for the power intensity of the incident wave, η stands for the vacuum impedance, and $\vec{E}$ and $\vec{H}$ represent the scattered electric field and magnetic field, respectively. $\hat{n}$ represents the unit vector of the external normal of the integral sphere, pointing to the outside of it, and is perpendicular to the surface. By normalizing the scattered light to the area of the SRR, one can determine the strength of the SRR’s response to the incident light. The calculated results of the normalized scattering cross-section can be found in Figure 2c, where a pronounced resonance peak at 2148 cm^−1^ is seen. Besides the blue shift, which can be easily seen by comparing the results in Figure 2c with those in Figure 2a, another significant difference is that the enhancement factor of the electric field amplitude is reduced to around 50 now (see Figure 2d). Since the local electric field is critical in achieving the strong coupling effect, one must be careful to achieve a judicious design for the SRR parameters.

We further fabricated the design structures and conducted some experimental investigations. We first performed a fine cleaning of CaF_2_ by using an ultrasonic bath first in acetone and then in ethanal to ensure that the no impurities and organic matters are remaining on the CaF_2_ surface. After a spin coating of PMMA (950 A4) at a precisely controlled rotation speed of 4000 rpm and subsequently baking the sample at 180 °C for 90 s, a uniform layer of about 200 nm thick PMMA was created. The sample was then loaded into a piece of electron-beam lithography (EBL) equipment (Raith Voyager, Dortmund, Germany) for the exposure. After developing, the sample was placed into the electron-beam evaporation (EBE) chamber. A vacuum condition below 3 × 10^−6^ Torr was used when the Ag was evaporated onto the sample at a coating rate of 0.5 Å/s to ensure the uniformity and quality of the coating. A final lift-off process in acetone helped to transfer the pattern in the electron-beam resist to the Ag layer, and the final arch-shaped SRR array structure was obtained. To reduce the influence of silver oxidation to the optical properties, the sample of a fabricated SRR array was loaded after fabrication into a vacuum type of Fourier transform infrared spectrometer (FTIR) (Bruker Vertex 80 V, Bruke Technology Co., LTD, Littleton, MA, USA) for the characterization of the transmission spectrum. The measurement was performed under normal incidence conditions with the help of a linear polarizer to make sure the incident line was *y*-polarized. The measured transmission spectrum is shown by the red solid line in Figure 3a, which exhibits a clear transmission dip at around 2028 cm^−1^. For comparison, we also presented using a black line in Figure 3a the result of transmission spectrum obtained from numerical simulations. It can be seen that a good agreement exists between experimental and theoretical results. The minimum measured transmittance of 0.3 is attributed to the additional optical losses stemming from manufacturing imperfections such as the unsmooth side walls of our SRR structure. Specifically, in the numerical simulations, the structural side wall was set to an ideal vertical state with no roughness. However, in the actual manufacturing process, some roughness may occur during the lift-off process. In addition, the slight oxidation of the silver material may also contribute to the slight discrepancy between the numerical and experimental results. Figure 3b displays a top-view Scanning Electron Microscope (SEM) image, revealing the fabricated quality of the SRR array.

In the next steps, we explore the strong coupling effect using PMMA thin layers with varying thicknesses, with the structure schematically shown in Figure 4a. To further understand the interaction between the SRR and the PMMA molecular vibration, one can use the classical coupled oscillator model (COM) for analysis, which provides us with an effective framework to explore the hybridization phenomenon between the MD resonance and PMMA molecular vibration [33].
(5)EB−iΓB2g gEM−iΓM2α1α2=E±α1α2

In this model, the energies of $E_{B}$ and $E_{M}$ correspond to the MD resonance of the SRR array and the PMMA vibration, respectively, whose dissipation losses are given by $\Gamma_{B}$ and $\Gamma_{M}$, respetively. The strong coupling between these two resonances will give rise to two polariton states $E_{\pm }$ given on the right side of Equation (5) as the eigenvalue of the new hybrid modes. These two states correspond to lower and upper polariton bands, which can be expressed as
(6)$$E_{\pm }=\frac{E_{B}+E_{M}}{2}\pm \frac{\sqrt{\left(E_{B}-E_{M}\right)+4g^{2}}}{2}$$

The dispersion relationship of the MD resonance, the PMMA molecular vibration, and the two new polariton states are shown in Figure 4b. The strong coupling leads to a distinct anti-crossover behavior with a Rabi splitting energy of 3.7 meV. The vibrational resonance of PMMA is represented by the horizontal dashed line, while the inclined white dashed line indicates the MD resonance. For simplicity, here we only consider the real part of the refractive index of the PMMA. The quite pronounced anti-crossing behavior is observable from the blue (upper and lower) dashed lines representing the two polariton modes, respectively. It is demonstrated that a good control over the PMMA-layer thickness provides an effective method to check the interaction between both resonances. The results obtained from the simulation calculations for three different thicknesses of PMMA coated on the SRR array are shown in Figure 4c. A strong coupling was evident for even the thickness of 80 nm, and a considerable spectral shift is observed for all the thicknesses, which shows that this structure has a high capability to detect organic material sensing. The spectral shift observed for the 80 nm thickness is 0.8 μm (from 2037 cm^−1^ to 1740 cm^−1^).

With the increase in the PMMA thickness, both polariton states show a redshift. However, when the PMMA thickness reaches about 130 nm, the red shift stops even with the further increase in PMMA thickness. This can be understood by the fact that the strong coupling mainly occurs in the gap on the right side of the SRR, whereas the height of the SRRs is only 50 nm. For the PMMA thickness beyond 130 nm, it can be taken as infinite relative to Ag height, so no obvious movement is observed in the transmission spectrum for a further increase in the PMMA-layer thickness. Furthermore, we experimentally verified the transmission spectrum measured by FTIR after coating with PMMA of different thicknesses on the structure, shown in Figure 4d. The results in Figure 4c,d confirm that experimental results are consistent with the numerical calculations, although the experimental results need slightly thicker PMMA layers to exactly match the simulation results. This discrepancy is due to the additional loss caused by the roughness of SRR side walls, which further reduce the overall Q factor and weakens the sensitivity of MD resonance to PMMA thickness. At the same time, the edge of the SRR array may result in an uneven and hard-to-control thickness of PMMA layer coated onto the sample. Even with those complications, the strong interaction between the MD resonance and PMMA vibration can be clearly distinguished.

## 3. Conclusions

In conclusion, by precisely adjusting the PMMA-layer thickness, we can accurately regulate the properties of the MD resonance and observe the mode splitting and anti-crossing behavior which is unique to the strong coupling phenomenon. This further shows that we can effectively adjust the strong coupling effect by fine-tuning the thickness of the PMMA thin layer. This finding not only demonstrates the complex and exquisite interactions between materials at a nanoscale, but also provides us with an effective means of optical sensing [34]. We expect that this research will promote the in-depth understanding of the interaction between matter and light in strong-coupling systems and promote the development of related applications.

## Figures and Tables

**Figure 1 sensors-24-02479-f001:**
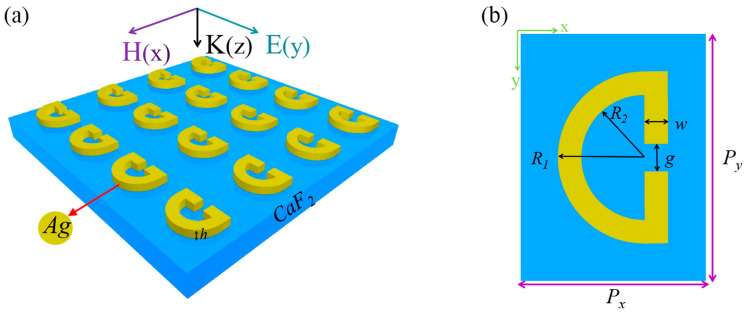
(**a**) illustrates a schematic representation of the structure consisting of an SRR array of silver material on a CaF_2_ substrate. (**b**) Shown is the overhead perspective of a single unit cell within the SRR array, in this work; *P_x_* = 1.53 μm, *P_y_* = 2.2 μm, *R*_1_ = 0.3*P_x_*, *R*_2_ = 0.185*P_x_*, *g* = 0.1*P_x_*, *w* = 0.1*P_x_.* By adjusting these parameters, the resonant frequency and resonance characteristics of SRR array can be regulated.

**Figure 2 sensors-24-02479-f002:**
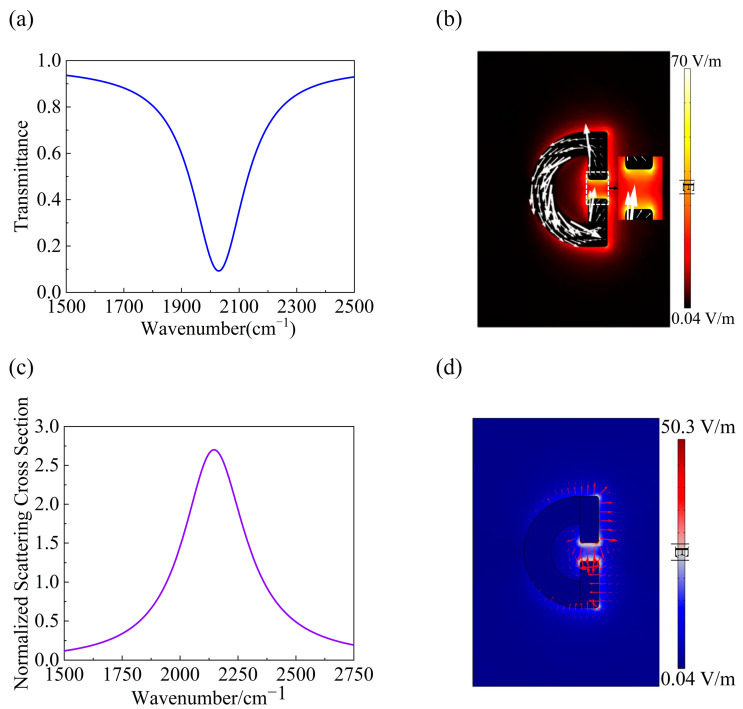
(**a**) Simulated transmission spectra through the metasurface supporting the MD resonance at *R*_2_ = 0.185*P_x_*. (**b**) Vectorial distribution of the displacement current across the *xy* plane overlapped with the amplitude of the electric field for *R*_2_ = 0.185*P_x_*. (**c**) Normalized scattering cross-section for *R*_2_ = 0.155*P_x_*. (**d**) Amplitude and vector distribution of the electric field in the SRRs structure across the *xy* cross-section for *R*_2_ = 0.155*P_x_.*

**Figure 3 sensors-24-02479-f003:**
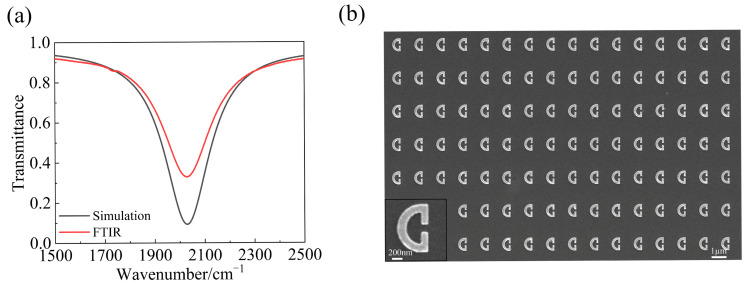
Diagram of the bare SRR transmittance and an SEM image. (**a**) The black solid line represents the simulated transmittance of the bare SRRs, whereas the red solid line depicts the experimentally determined transmittance of the same. (**b**) Presented is an SEM image, exhibiting the overhead perspective of the fabricated bare SRR array.

**Figure 4 sensors-24-02479-f004:**
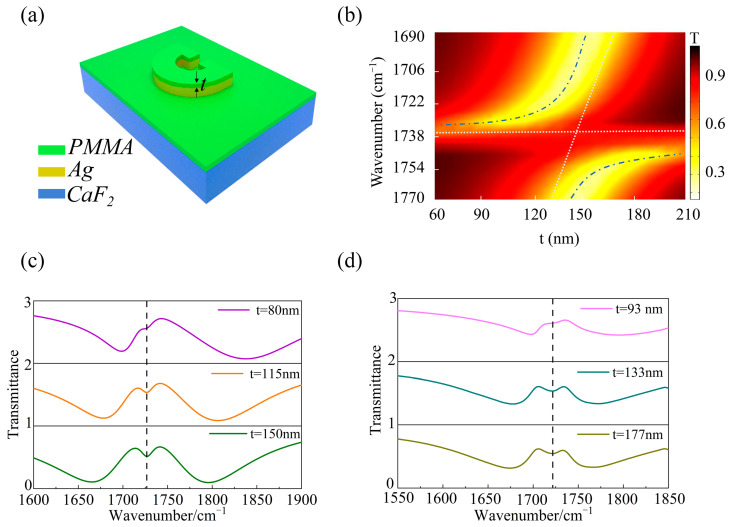
Experimental characterization and theoretical simulation of the strong coupling effect at different thicknesses of PMMA films. (**a**) Schematic setup of the model showing the strong coupling phenomenon observed at the presence of PMMA films. (**b**) The transmission spectrum as a function of the coated PMMA thickness. (**c**) The calculated transmission spectrum at normal incidence, considering three different thicknesses of the PMMA layer (*t* = 80 nm, 115 nm, and 150 nm). (**d**) Transmission spectra for different thicknesses of PMMA layers spin-coated on the SRRs structure, measured using FTIR. The vibrational resonance positions of PMMA molecules are marked by the black vertical dashed lines in (**c**,**d**).

## Data Availability

The original contributions presented in the study are included in the article, further inquiries can be directed to the corresponding author.

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
