# Peer review of "The Strong Coupling Effect between Metallic Split-Ring Resonators and Molecular Vibrations in Polymethyl Methacrylate"

_sensors, 2024, doi:10.3390/s24082479_

Round 1

Reviewer 1 Report

Comments and Suggestions for Authors

Comments on the manuscript

The manuscript investigates the strong coupling effect between metallic split ring resonators (SRRs) and molecular vibrations in polymethyl methacrylate (PMMA). Through theoretical simulations and experimental fabrication/characterizations, the study explores the interaction between the magnetic dipole resonance of the SRRs and the vibrational resonance of PMMA molecules. By adjusting the parameters of the hybrid SRR array, such as PMMA thickness, the resonant frequencies can be controlled to achieve strong coupling between modes (anti-crossing) in the transmission.

In my assessment, the experimental/theoretical findings of this work contribute to the metamaterials-enabled light-matter interaction in mid-IR, a topic of high interest. Hence, I support the publication of this work, provided that certain concerns are addressed. Please find my suggestions and comments for the author below.

“Therefore, such resonators are extensively used in the field of micro and nano optics serving as absorbers[8][9] and filters [10][11] etc.” Correct. However, it needs more recent works on absorbers/filters to support this claim. For example [Kivshar et al. "Bound states in the continuum in anisotropic plasmonic metasurfaces." Nano Letters (2020); Thomas Feurer et al. "Ultrafast and low-threshold THz mode switching of two-dimensional nonlinear metamaterials." Nano letters (2022)].

About the PMMA layer thickness control issue, especially about the imprecise control over the thickness and the presence of roughness on the SRR sidewalls, how did these differences between real experiments and simulation modeling impact the strong coupling effects?

The authors claim magnetic resonance mode in their SPR arrays without running the multipolar decomposition. Not sure if it is right. Also, I am not sure if the magnetic dipolar resonance is important for the Mid-IR spectral enhancement. Please justify.

Also, silver is not stable when they are exposed to the air environment. How did you make sure of the stability of your fabricated metasurfaces during the whole experiment?

How did the authors understand the discrepancies between the simulation results against the experimental data (Fig. 3a)? Discussing the difference between simulation and experimental results, such as the amplitude difference, would strengthen the credibility of your findings.

Comments on the Quality of English Language

readable

Reviewer 2 Report

Comments and Suggestions for Authors

The authors present a study on the strong coupling between metamaterials and the vibrational resonances of chemical bonds in polymer molecules utilizing split-ring-resonators (SRR). The strong coupling between the plasmon resonance and vibrational resonance of molecules in the MIR band that is highly sensitive to the thickness of the introduced PMMA layer is numerically investigated using FEM. Moreover, an experimental validation is given. The results are interesting and can be useful for future studies of the strong coupling phenomenon as well as for the design of nanoscale structures to achieve organic material sensing.

However, before a final decision, a few points should be clarified to improve the paper:

1) The investigation considers only normal incidence. It should be interesting for the reader to see results on transmission in the case of oblique incidence and how the resonance wavelength is affected.

2) Please define the normalized scattering cross-section presented in Fig. 2c and describe shortly its meaning and the method of its calculation.  

3) A diagram (or contour) showing the resonant wavelength versus different PMMA thicknesses (t) could be given for completeness.

4) Reference [20] is incomplete. Please check the source.

5) The authors may give a specific example of the application of their study to organic material sensing in the conclusions.  

6) The authors should read carefully the submitted manuscript and correct some minor ‘English language’ syntactic errors.

Comments on the Quality of English Language

The authors should read carefully the submitted manuscript and correct some minor ‘English language’ syntactic errors.

Reviewer 3 Report

Comments and Suggestions for Authors

This manuscript, "The strong coupling effect between metallic split ring resonators and molecular vibrations in polymethyl methacrylate," presents an interesting study on the interaction between metamaterials and vibrational resonances of polymer molecules. The authors propose to utilize metallic split-ring resonators (SRRs) to explore the strong coupling phenomenon on the nanoscale, which is a topic of significant importance in the field of plasmonics and metamaterials.

Overall, the manuscript is well-structured and provides a clear description of the experimental setup, numerical simulations, and results. However, there are several points that need to be addressed for further improvement.

1.     the Abstract section should be concise and provide a brief overview of the study. While the current Abstract covers the main points, it could be shortened and more focused on the key findings and implications of the work.

2.     The Introduction section should provide a more comprehensive background on the topic, including a brief review of previous studies on strong coupling between metamaterials and molecular vibrations. This would help the reader better understand the significance and novelty of the current study.

3.     The manuscript could benefit from a more detailed description of the experimental methods and numerical simulations. Specifically, it would be helpful to provide additional information on the design and fabrication of the SRRs, as well as the specific techniques used for characterizing the strong coupling effect.

4.     The Discussion section could be expanded to include a more thorough analysis of the results. The authors should discuss the observed anti-crossing behaviors and spectral splitting in greater detail, explaining their significance and potential applications.

5.     The manuscript should be carefully proofread to eliminate any grammatical errors or typos. Attention should also be paid to the consistency of notation and terminology throughout the text.

6.     The author should supplement the relevant calculations of pattern coupling in Figure 4 (b), such as supplementing relevant coupled oscillator model to help understand coupling behavior, and also provide important basis for implementing strong coupling. The specific calculation can refer to [Physical Review A 109, 013504 (2024)].

7.      There are some highly related references for strong coupling missing, for example, [Reports on Progress in Physics, 78(1): 013901, 2014], [Carbon 145, 596-602, 2019], [Nature, 535(7610): 127-130, 2016], [Nature communications, 2020, 11(1): 2715.], [ACS Photonics 6 (11), 2884-2893, 2019].

In conclusion, this manuscript presents an interesting study on the strong coupling effect between metallic split ring resonators and molecular vibrations in polymethyl methacrylate. While the work shows promise, it could be improved by addressing the points mentioned above. With these revisions, the manuscript would be a valuable contribution to the field of plasmonics and metamaterials.

Round 2

Reviewer 2 Report

Comments and Suggestions for Authors

The authors have responded adequately to my recommendations and comments. The revised manuscript has been improved a lot and deserves publication in 'Sensors' Journal.  

Reviewer 3 Report

Comments and Suggestions for Authors The careful editing of the manuscript improves its quality. I believe the manuscript is ready to be published as is.